# Determining the Degree of [001] Preferred Growth of Ni(OH)_2_ Nanoplates

**DOI:** 10.3390/nano8120991

**Published:** 2018-11-30

**Authors:** Taotao Li, Ning Dang, Wanggang Zhang, Wei Liang, Fuqian Yang

**Affiliations:** 1College of Materials Science and Engineering; Shanxi Key Laboratory of Advanced Magnesium-based Materials, Taiyuan University of Technology, Taiyuan, 030024, China; xueyanles10@126.com (T.L.); zwgang0117@163.com (W.Z.); 2Laboratoire de Chimie Physique et Microbiologie pour les Matériaux et l′Environnement (LCPME), UMR 7564, CNRS-Université de Lorraine, Villers-lès-Nancy 54600, France; ning.dang@univ-lorraine.fr; 3Materials Program, Department of Chemical and Materials Engineering, University of Kentucky, Lexington, KY 40506, USA

**Keywords:** degree of preferred growth, Ni(OH)_2_ nanoplate, shape function

## Abstract

Determining the degree of preferred growth of low-dimensional materials is of practical importance for the improvement of the synthesis methods and applications of low-dimensional materials. In this work, three different methods are used to analyze the degree of preferred growth of the Ni(OH)_2_ nanoplates synthesized without the use of a complex anion. The results suggest that the preferred growth degree of the Ni(OH)_2_ nanoplates calculated by the March parameter and the expression given by Zolotoyabko, which are based on the analysis and texture refinement of the X-ray diffraction pattern, are in good accordance with the results measured by SEM and TEM imaging. The method using the shape function of crystallites is not suitable for the determination of the preferred growth degree of the Ni(OH)_2_ nanoplates. The method using the March parameter and the expression given by Zolotoyabko can be extended to the analysis of block materials.

## 1. Introduction

All polycrystalline materials, including block and powder materials, exhibit preferred orientation of crystallites, to some degree, i.e., texture, due to the mechanical deformation and microstructure evolution during material processing [1,2,3]. It is of great importance to analyze and determine the preferred orientation when the material properties are orientation-dependent. Electrical, magnetic and mechanical properties of crystals with cubic symmetry are independent on the orientation [4,5,6].

There are several techniques available to determine the degree of the preferred orientation of powder materials, including optical microscopy (OM), scanning electron microscopy (SEM) and transmission electron microscopy (TEM). The principle in TEM using to determine the preferred growth degree of powder materials is based on the analysis of TEM images and the corresponding diffraction pattern or phase-contrast images along appropriate crystal axis [7]. However, both SEM and TEM can only reveal limited portion of the preferred growth of powder materials, and OM can only be used to analyze the structures with characteristic dimensions more than 200 nm. Also, it is easy to introduce artifacts in replicating real 3-D morphology from projected 2-D images from optical/electronic images, especially TEM images. It is very difficult to use the imaging-related techniques to determine the degree of the preferred growth of nanostructures.

X-ray diffraction (XRD) is another effective way to determine the preferred orientation degree in polycrystalline materials. With a tilting holder in the X-ray diffractometer, pole figure/inverse pole figure or the orientation distribution function can be used to characterize the blocking materials, such as metals. Yet, there are limited reports on the XRD use in the analysis of the preferred growth degree of powder materials, such as nanostructured Ni(OH)_2_ and Cd(OH)_2_ with hexagonal or trigonal symmetry, which have specific preferred growth of [001] [8,9]. In this study, we synthesize Ni(OH)_2_ nanoplates without the use of any complex anion, and calculate the March parameter (r_n_) of the preferred growth with fitting the XRD spectra of the Ni(OH)_2_ nanoplates by the whole powder pattern fitting method (WPPF) in the framework of the March-Dollase function [10,11]. Afterwards, the degree of the preferred growth degree (η) of the Ni(OH)_2_ nanoplates is determined with using Zolotoyabko’s normalized equation (η_X_) [1], the ellipsoidal model (η_E_) and direct measurement (η_M_) in SEM and TEM images. The preferred growth degree within the ellipsoidal model and direct measurement in SEM and TEM is defined by the ratio of the short axis and long axis in average (η = D_00l_/D_hk0_). The overall aim in this study is to offer a simpler way to determine the degree in the preferred powder materials.

## 2. Experimental Section

### 2.1. Fabrication of Ni(OH)_2_ Nanoplates

NiCl_2_·6H_2_O (99%) and NaOH (99%) were obtained from Sigma-Aldrich (Sinopharm Chemical Reagent Co., Ltd, Shanghai, China) without further purification. The as-received 0.1 M NiCl_2_·6H_2_O of X mL and 0.2 M NaOH of 2X mL were mixed under magnetic stirring to form a homogeneous solution (solvent: deionized water). Multiple centrifugations were used to remove the sodium chloride precipitates from the solution. The homogeneous solution of 40 mL was placed in a Teflon-lined stainless-steel autoclave (Yanzheng experimental instrument co., LTD, Shanghai, China, 60 mL), which was sealed and maintained at 180 °C for 10 h and cooled to room temperature in the furnace. The light green product collected from the Teflon-lined stainless-steel autoclave was washed by distilled water and ethanol. The washed product was dried in a vacuum oven at 60 °C for 6 h.

### 2.2. Materials Characterization

The XRD analysis of the synthesized products (Ni(OH)_2_ nanoplates) was performed on a Rigaku X-ray diffractometer (40 kV, 40 mA, Cu Kα radiation: 1.54184 Å, Rigaku Corporation, Akishima-shi, Tokyo, Japan) equipped with a one-dimensional array detector (DteX250(H) Rigaku Corporation, Akishima-shi, Tokyo, Japan) at room temperature. The incident Sola slit and the length of limiting slit are 1/6° and 10 mm, and the scan step is 0.01° for the 2θ in a range of 10° to 100°. The morphology and the electron diffraction pattern of the synthesized products (Ni(OH)_2_ nanoplates) were analyzed on a scanning electron microscope (MIRA3 LMH, TESCAN Corporation, Brno, Czech) and transmission electron microscope (JEOL 2100F, JEOL ltd, Akishima, Tokyo, Japan), respectively.

### 2.3. Methodology

The analysis of the XRD patterns was performed, using the software of Rigaku SmartLab Studio II (Rigaku Corporation, Akishima-shi, Tokyo, Japan). The second derivative method was used to identify the diffraction peaks in the XRD patterns, and the split pseudo-Voigt function was used to fit a single peak. The whole powder pattern fitting method (WPPF) without reference to a structural model, as proposed by Pawley in the analysis of the neutron powder data, was used to fit the diffraction spectra [12]. The texture refinement was used to determine the March parameter from the March-Dollase function (W(α)) [10,11](1)$$W(\alpha )={\left(r_{n}^{2}\cos^{2}\alpha_{n,h}+r_{n}^{-1}\sin^{2}\alpha_{n,h}\right)}^{-3/2}$$
here, α_n,h_ is the angle between the orientation vector and diffraction plane vector. The March number r_n_ determines the preferred orientation strength. The March-Dollase function represents the crystallite fraction with the reciprocal lattice vectors being perpendicular to the sample surface [1]. For r_n_ = 1, there is no preferred orientation (random orientation); for r_n_ < 1, there is a preferred orientation by plate crystallites with the orientation vector perpendicular to the plate surface; and for r_n_ > 1, there is a preferred orientation by needle crystallites with the orientation vector parallel to the longitudinal direction of needles.

## 3. Results and Discussion

In Figure 1A, the XRD pattern of the synthesized product is shown, in which all the diffraction peaks can be indexed by the trigonal structure of theophrastite (Ni(OH)_2_ mineral) with the space group P-3m1 [13]. However, comparing with the peaks calculated from the crystallographic information file (ICSD 24015), as shown by the calculated XRD pattern, a slight offset is observed in the diffraction angles (2θ) between measured data and calculated results, which likely is due to the difference in the lattice constants. After performing the refinement calculation of the lattice constants (a, b and c) and the profile function parameters, we obtain lattice constants of the synthesized Ni(OH)_2_: a = 3.13212(11) and c = 4.6102(4) Å.

Figure 1A also depicts the calculated XRD pattern of the synthesized Ni(OH)_2_ with indexed planes. The modified crystallographic file was used in the further profile and texture refinement to determine the preferred direction degree.

It is known that diffraction peak experiences an angle broadening in crystalline materials with grain size less than 100 nm or under lattice strain [14]. The diffraction peak broadening for the synthesized Ni(OH)_2_ is likely due to the submicron crystallite sizes. From Figure 1A, we note that the FWHM (full width at half maximum) of the (001) diffraction peak of Ni(OH)_2_ is much wider than those of (100) and (110), suggesting that the crystallite size along the [001] direction is much smaller than those along the [100] and [110] directions and the crystallites of Ni(OH)_2_ are ellipsoidal instead of spherical shape.

To separate the instrumental broadening from the peak broadening of Ni(OH)_2_, we used a silicon standard specimen for the instrumental broadening estimation during the comprehensive analysis of the synthesized Ni(OH)_2_ by SmartLab Studio II. Table 1 lists the corrected FWHM values of the synthesized Ni(OH)_2_. The FWHM of the peaks (001), (002), (003) and (004) are 0.395, 0.54, 0.43 and 0.33°, respectively, which are much larger than those of the other peaks; the FWHM of the peaks (100), (110) and (200) are 0.083, 0.143 and 0.09°, respectively. The analysis of the widest peak (00l) and the narrowest peak (hk0) parameters suggests that the synthesized Ni(OH)_2_ crystallites are two-dimensional in shape, and the diameter of synthesized Ni(OH)_2_ crystallites along [hk0], D_hk0_, is much larger than that along the [00l] direction, D_00l_. Thus, the synthesized Ni(OH)_2_ is present in the two-dimension nanoplate shapes.

In the standard diffraction spectra calculated with spherical crystallites, the integrated intensity of (101) should be larger than that of (001), as shown in Figure 1B (sphere). According to the measured diffraction data, there exists the preferred growth of Ni(OH)_2_ crystallites responsible for the abnormal integrated intensity ratio between (001) and (101). Using the March-Dollase function in the smartLab Studio II, we performed the profile and texture refinement to determine the degree of the preferred direction [001], i.e., the smallest dimension of the two-dimensional Ni(OH)_2_ crystallites. The calculation gives the March parameter of r_001_ = 0.8113, as shown in the calculated XRD spectrum of Figure 1A with R_wp_ = 8.08% and S = 0.9925.

Figure 1B depicts the variation of the simulated diffraction spectra with the March parameter r_00l_. Increasing the March parameter r_00l_ leads to the increase of the intensity ratio between (001) and (101). This ratio likely represents the preferred growth degree of the synthesized Ni(OH)_2_. The standard theophrastite (JCPDS. No.14-0117) given by ICDD also shows preferred growth because the ratio of the diffraction intensity of (001) and (101) is 1. Yet, the March parameter r_00l_ cannot effectively represent the preferred growth degree of the 2-D Ni(OH)_2_ crystallites.

Zolotoyabko established a relationship between the degree of preferred growth and the March parameter r_n_ extracted from diffraction measurements using the nominalized March-Dollase function W(α). He proposed that the degree of preferred growth can be expressed by η as(2)$$\eta =100\%{\left[\frac{{\left(1-r_{n}\right)}^{3}}{1-r_{n}^{3}}\right]}^{1/2}$$
for r_001_ = 0.8113, we obtain the degree of the [001] preferred growth deduced in X-ray diffraction, η_X_, as 12.00%.

Žunić and Dohrup demonstrated that an ellipsoid in reference to crystallographic axes (triclinic case) can be expressed as [15](3)$$b_{11}x_{c}^{2}+b_{22}y_{c}^{2}+b_{33}z_{x}^{2}+2b_{12}x_{c}y_{c}+2b_{13}x_{c}z_{c}+2b_{23}y_{c}z_{c}=1$$
where b_ij_ (i, j = 1, 2, 3) are the coefficients related to reciprocal axes. The symbols x_c_, y_c_ and z_c_ stand for the parameters in describing the surface function. Thus, the simulated ellipsoid can be expressed as(4)$$b_{11}h^{2}+b_{22}k^{2}+b_{33}l^{2}+2b_{12}hk+2b_{13}hl+2b_{23}kl=\frac{9}{16D_{hkl}^{2}d_{hkl}^{2}}$$
here, D_hkl_ is the surface average diameter of the ellipsoidal crystallites (D_00l_ and D_hk0_) obtained from the normalized expressions of anisotropic line broadening, and d_hkl_ is the interplanar spacing. The meanings of the letters: h, k and l are consistent with the subscript in D_hkl_ and d_hkl_. Table 2 lists the coefficients of the refined ellipsoidal crystallites of the synthesized Ni(OH)_2_. Using the fitting coefficients for the synthesized Ni(OH)_2_, we obtain the main diameters of D_00l_ and D_hk0_ as 31.40 and 185.95 nm, respectively. The synthesized Ni(OH)_2_ indeed is present in the shape of nanoplates with the [001] preferred growth and the degree (η_E_) of the [001] preferred growth is 16.89%.

Figure 2A,B,D,E shows the SEM and TEM images of the synthesized Ni(OH)_2_, confirming the shape of nanoplates determined by the analysis of the XRD pattern via the whole powder pattern. Figure 2C depicts the selected area electron diffraction (SAED) pattern of the Ni(OH)_2_ nanoplates. If the diameter of a grain in the preferred direction is much longer or shorter than that of its normal directions, it can be rendered as the preferred growth. Thus, the Ni(OH)_2_ nanoplates exhibit six-fold symmetry about the [001] direction in accordance with the preferred growth obtained by X-ray diffraction. The synthesized Ni(OH)_2_ powder contains two-dimension nanoplates with the normal of the surface of the nanoplates being (001) and D_001_ being much smaller than D_hk0_. To confirm the trigonal symmetry, in Figure 2C, the SAED pattern is shown for a single crystal Ni(OH)_2_ nanoplate. Thus, the degree of the preferred growth of Ni(OH)_2_ grains characterized by the X-ray diffraction could be revealed by the Ni(OH)_2_ nanoplates observation by SEM and TEM. It needs to point out that the phase-contrast images may not reveal the structure of the Ni(OH)_2_ nanoplates due to possible damage induced by electron beam with an energy of 200 KeV.

Figure 2D,E shows the SEM and TEM images of the side surfaces of the synthesized Ni(OH)_2_ nanoplates. The SAED pattern of the square area in Figure 2E is shown in Figure 2F. The one-way diffraction spots revel the spacing of [001] for the Ni(OH)_2_ nanoplates in consistence with the plane (001) normal of the surface of the nanoplates. Note that the crystal orientation cannot be indexed by the one-way diffraction pattern.

With using an ellipsoid model, Žunić and Dohrup proved that the average ellipsoid size follows an equivalent ellipsoidal function in X-ray diffraction [15]. As follows from Equation 1, the orientation distribution depends only on March parameter (r) and angle (θ), with no crystallite size being involved. However, the degree of the preferred Ni(OH)_2_ nanoplates could be determined from measuring the edge length of the preferred growth and its normal diameter. Thus, the preferred direction and normal direction diameters are measured to confirm the results in X-ray diffraction. In this paper, the degree of averaged crystallite shape in [001] preferred Ni(OH)_2_ is simply defined as the ratio of D_00l_ and D_hk0_ labelled in Figure 2F. From Figure 2D,E, we note that there are some Ni(OH)_2_ nanoplates tilted or truncated, and expect to obtain a distribution of the diameters of D_hk0_ and D_00l_. Statistical analysis of the SEM and TEM images of the side surfaces of 288 Ni(OH)_2_ nanoplates was performed in Image J to calculate the diameters of D_hk0_ and D_00l_. We obtain D_00l_ of 24.34 ± 3.54 nm.

Figure 3 shows the distribution of D_hk0_ of the Ni(OH)_2_ nanoplates over 288 nanoplates. The D_hk0_ is in the range of 129.31–539.68 nm, and the averaged D_hk0_ is 278.4 nm. Using the D_hk0_ and D_00l_, we obtain the degree of [001] preferred growth as 8.74%. The λ-like distribution of the D_hk0_ of the Ni(OH)_2_ nanoplates, as shown in Figure 3, reveals the difficulty in the counting of Ni(OH)_2_ nanoplates of small diameters, resulting in smaller η_M_ than η_X_. We conclude that the degree of the [001] preferred growth of the Ni(OH)_2_ nanoplates, η_X_, calculated by the WPPF reflects the degree of the [001] preferred growth of the Ni(OH)_2_ nanoplates, η_M_.

Compared the values of 278.40 and 24.34 nm of D_hk0_ and D_00l_ measured by SEM and TEM with those of 185.95 and 31.40 nm calculated from the ellipsoidal model, we note that there are significant differences between the corresponding values. The shape function of crystallites is not suitable for the description of the shape of nanoplates, and the calculated values cannot be used to represent diameters of D_hk0_ and D_00l_ of the Ni(OH)_2_ nanoplates.

For the situation when the shape parameters of crystallites cannot be accounted for the actual sizes, an average spherical shape can be assumed. Using the Halder-Wagner’s method, we can calculate D_hk0_ from the slope of the line plotting of $\beta^{2}/\tan^{2}\theta$ vs. β/tanθsinθ for the FWHM of (100), (110)and (200) and D_00l_ from the FWHM of (001), (002), (003) and (004) [16]. As shown in Appendix A the calculated diameters of D_00l_ and D_hk0_ are 22.2 ± 1.3 and 93.7 ± 14.1 nm, respectively. The edge length in [001] of the Ni(OH)_2_ is in good relation to the results 24.34 nm determined by SEM and TEM images. The diameter of the crystallites below 100 nm will result the widening in the diffraction peaks, which may be accounted for the inappropriate edge length in [hk0] [14].

## 4. Summary

We have synthesized Ni(OH)_2_ nanoplates without the use of any complex anion. Three different methods are used to analyze the degree of preferred growth of the Ni(OH)_2_ nanoplates. Using the March-Dollase method and the whole powder pattern fitting method, we obtain the March parameter of r_001_ = 0.8113 through the texture refinement. The degree of [001] preferred growth of the Ni (OH)_2_ nanoplates, η_X_, is found to be 12.00% from the March parameter of r_001_ and the normalized expression given by Zolotoyabko. The shape function of crystallites is not suitable to directly describe the shape of the Ni(OH)_2_ nanoplates. Using the Halder-Wagner’s method, we obtain the edge length of D_001_ as 22.2 ± 1.3 nm that is in good relation to the results measured by SEM and TEM imaging. The TEM imaging of the Ni(OH)_2_ nanoplate reveals that the Ni(OH)_2_ nanoplates exhibit six-fold symmetry about the [001] being the preferred growth. The measured degree of [001] preferred growth of the Ni(OH)_2_ nanoplates, ηM, is 8.74%, slightly less than ηx. The March parameter and the expression given by Zolotoyabko can be used to effectively calculate the degree of preferred growth of nanostructural materials (0 < r < 1), which can be extended to the analysis of block materials.

## Figures and Tables

**Figure 1 nanomaterials-08-00991-f001:**
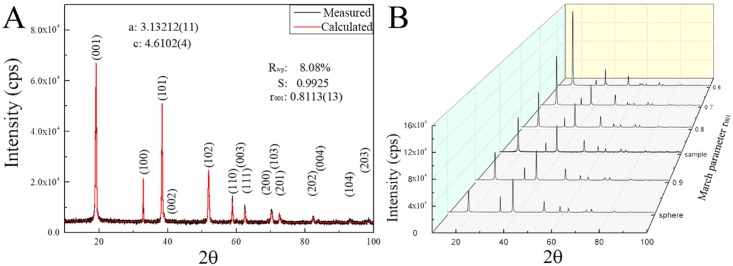
(**A**) The measured and simulated XRD spectra (the smaller the value of R_wp_, or the closer the value of S to 1, the better is the match between the measured and simulated data), and (**B**) the simulated XRD diffraction spectra with strong preferred orientation of the [001] type (r_n_ = 1 (sphere) without preferred orientation; r_n_ < 1 with the preferred orientation by plate crystallites; and r_n_ > 1 with the preferred orientation by needle crystallites).

**Figure 2 nanomaterials-08-00991-f002:**
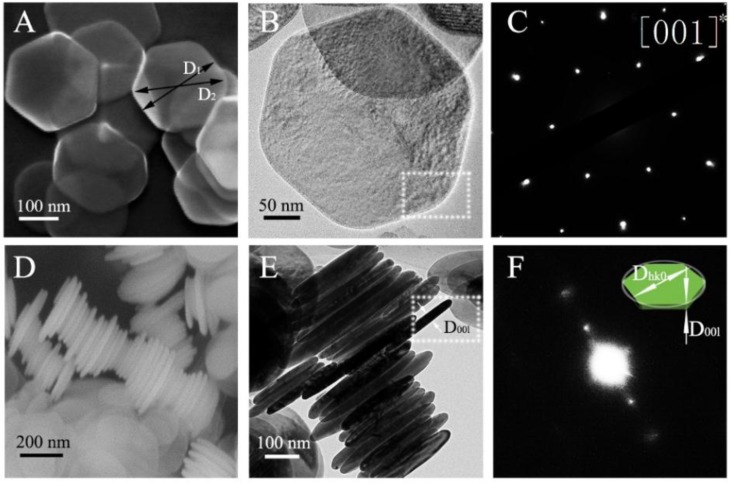
(**A**) SEM and (**B**) TEM images of Ni(OH)_2_ nanoplates (top view), (**C**) SAED pattern of the square area in B, (**D**) SEM and (**E**) TEM mages of Ni(OH)_2_ nanoplates (side view), and (**F**) SAED pattern of the square area in E (The inset in F is a schematic of a Ni(OH)_2_ nanoplate showing the diameter of D_hk0_ and D_00l_).

**Figure 3 nanomaterials-08-00991-f003:**
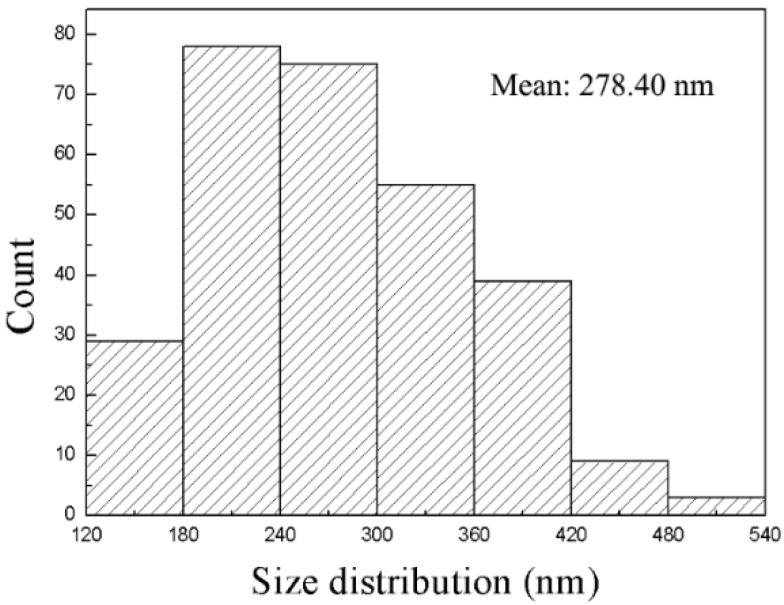
Distribution of D_hk0_ of the Ni(OH)_2_ nanoplates over 288 nanoplates.

**Table 1 nanomaterials-08-00991-t001:** FWHM (β) of the measured peaks of Ni(OH)_2_ after the correction of the instrumental broadening by external standard method (Si) from Rigaku.

2θ (°)	Int. W (°)	hkl	2θ (°)	Int. W (°)	hkl
19.147	0.395	001	69.201	0.09	200
32.877	0.083	100	70.24	0.53	103
38.380	0.235	101	72.545	0.40	201
38.92	0.54	002	82.43	0.38	202
51.920	0.38	102	83.91	0.33	004
58.860	0.143	110	93.10	0.46	104
60.22	0.43	003	98.45	0.42	203
62.529	0.22	111			

**Table 2 nanomaterials-08-00991-t002:** Coefficients of the ellipsoid in simulating the preferred growth of Ni(OH)_2_ (the ellipsoid is characterized by six coefficients).

b_11_	b_22_	b_33_	b_12_	b_13_	b_23_
7.029E-8	7.029E-8	8.506E-7	3.515E-8	0	0

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
