# Peer review of "Determining the Degree of [001] Preferred Growth of Ni(OH)_2_ Nanoplates"

_nanomaterials, 2018, doi:10.3390/nano8120991_

Round 1
Reviewer 1 Report
My review is in the attached file.

Author Response
I am much obligated for your time to improve this manuscript. The detailed responses have been uploaded as attachments.

Reviewer 2 Report
The paper reports the comparison of determination methods of the degree of preferred orientation using Ni(OH)2 nanoparticles. I think the paper contains some leaps in logic and misdirected comparisons, which are the reason I cannot recommend the paper for publication. Please find the detailed comments below:
i) The authors explained that the paper compared three different methods for analyzing the degree of preferred orientation. a) Zolotoyabko’s method using March-Dollase function–this is the degree of preferred orientation obtained from XRD peak intensities. This represents the preferred orientation in the powder packed in the XRD sample holder. b) Žunić’s method–this is the crystallite size estimation along (hk0) and (001) axes, Dhk0 and D001, from the inhomogeneous broadenings of the peaks in an assumption of ellipsoid crystallites. c) Direct measurement of the crystallite sizes from TEM images. In b,c), the authors simply defined the preferred orientation as the ratio of short/long crystallite sizes, D001/Dhk0. This represents the preferred orientation of each crystallite, in other words, the anisotropic shape of the crystal. It is true that the crystal shape affects the tendency of the preferred orientation in the powder in the holder, but, direct comparison between the degree of orientation and crystallite size ratio would be nonsense.
ii) The nanosized plate-like particles tend to form a stacking aggregate, as actually shown in Fig. 2D,E, and act as one large particle in powder. You cannot discuss the preferred orientation in powder without specific information on the aggregate structure of primary nanocrystals.
iii) In my opinion, some methodologies should be explained in more detail. I did not totally understand how the authors obtained bij values (Table 2) in Žunić’s method.
Author Response
I am much obligated for your time to improve this manuscript. On the basis of your opinion, detailed responses have been uploaded as attachment.

Reviewer 3 Report
In the paper by Li et al (Nanomaterials-379598), the authors analyze XRD data to determine the degree of the preferred orientation in polycrystalline Ni(OH)2 and compare the results with those inferred by TEM and SEM measurements.
In a first reading the manuscript seems convincing. However, after careful attention, deficiencies, aspects without specifying, and bad referenced citations are detected, leading to errors in the information given to the reader. Also, the writing in English is poor. It must be reviewed by a native to avoid expressions like the following (they are only a couple of examples):
In Abstract: “In this work, we use three different methods are used …”
In Introduction: “The principle in using TEM determine the degree of the preferred orientation of powder materials is based on …”
In Results: “The standard theophrasite (JCPDS. No. 14-0117) given by ICDD also shows preferred orientation for the ratio of the diffraction intensity of (001) and (101) is 1”
After equations 1, 2 and 4 here, for and here, respectively, must be written in uppercase.
The publication year of the reference 7 is 2008, not 2010.
Regarding to the Introduction: 1) at the beginning of it (lines 29 and 30), the authors should minimally specify the importance of preferential orientation in the materials properties, including appropriate references, in order to highlight the interest of their work. 2) At the end of this section, where only the purpose and the significance of the work, and the current state of the research field should be addressed, some results are stated. Moreover, the authors cite the Emil’s equation, which is not referred to elsewhere in the article. In my opinion, the guidelines of the different methods of analysis (advanced in the Abstract) should be anticipated at this point.
In Materials Characterization, authors must indicate the wavelength of the X-ray source and the temperature at which the diffraction experiments were carried out. Are the diffraction data deposited in any database?
At the beginning of Results and Discussion, the authors say that the diffraction peaks are indexed by the trigonal structure of theophrastite, a very important point for the article, giving the reference 12. This reference corresponds to a paper concerning with the determination of enzyme dissociation constants!!! Its correction is necessary.
Ni(OH)2 is not an appropriate entrance for Table 1. It may be replaced by (hkl).
When writing an equation, the meaning of all its symbols is usually indicated. This is not the case with equations 3 and 4. On the other hand, It is not straightforward the deduction of the degree hE of the [001] preferred orientation for the synthesized Ni(OH)2, and authors must explain this point. With this ellipsoidal analysis, they obtain values for D001 and Dhk0. A later comparison with the values measured by SEM and TEM reveals significant differences. Equation 3 has been proposed for the triclinic case. I wonder if it will be applicable to the trigonal case corresponding to the synthesized Ni(OH)2. Could this fact be related to the differences indicated above?
The authors conclude: “The shape function of crystallites is not suitable for the description of the shape of nanoplates, and the calculated values cannot be used to represent diameters of …nanoplates”, and then they assume an average spherical shape and make use of the Halder-Wagner’s method through the integral breadths plotting b2/tan2q vs. b/tanqsinq. Three considerations about this strategy: 1) Or I am misunderstanding or it contradicts the previous evidences, particularly those of TEM and SEM. A clear explanation is required. 2) Again, the meaning of the terms of the previous equation must be given in the text. 3) The incorporation of the corresponding graph in the manuscript would be highly desirable.
Finally, on page 4 line 37 the acronym SAED is mentioned for the first time in the manuscript, which refers to a selected area in electron microscopy. The authors must explicitly indicate in the text the meaning of this acronym.
The authors must correct the indicated points and clarify these questions so that the paper can be published with the standards of Nanomaterials journal.
Author Response

(The authors gave the same response as above.)

Round 2
Reviewer 1 Report
After the revision, the paper is noticeably improved and, in my opinion, it can be published in the present state.
Reviewer 2 Report
It seems the revised manuscript does not reflect my previous comments. The essence of the comments was: do not mix up the preferred orientation and the preferred growth of crystallites.
Preferred orientation is defined in polycrystalline sample, in this case, the whole powder packed in the XRD sample holder. Preferred orientation represents the fraction of nanoparticles packed along a specific crystallographic direction in the powder. This was obtained from the fitting using March-Dollase function. This is related to the peak intensity ratios of the XRD profile.
Preferred growth of crystallites can be defined in each crystallite, in the present case, each nanoparticle. In the present paper, Žunić’s method, direct observation in TEM, and Halder-Wagner plot give the average preferred growth of crystallites, in other words, the average shape of anisotropically grown nanoparticles. This is related to the peak widths of the XRD profile.
The previous comment i) said direct comparison between the preferred orientation and preferred growth of crystallites would be nonsense. The previous comment ii) said the same item: the direction of preferred orientation in the powder could be totally different from the preferred growth direction of crystallites because of the aggregation.
Author Response
Thanks for your review. The evaluation is very professional for improving this manuscipt. The revised part is marked in blue.

Reviewer 3 Report
Both in the second version of the manuscript and in the response letter, most of my suggestions and questions have been addressed by the authors.
However, in my opinion, a correction of English is still necessary. For example, the first sentence highlighted in red of the new Introduction of the article, “The physical properties, such as electrical property and magnetic properties, and mechanical property in the crystal of cubic symmetry, are independent on orientation“, should be written more appropriately as: Electrical, magnetic and mechanical properties of crystals with cubic symmetry are independent on the orientation.
The acronym ODF is not necessary since it is not used again throughout the manuscript.
As I had expressed in my previous report, the Introduction should not refer to results of the work. Therefore, in the sentence “Using Zolototabko’s normalized equation [1], we determine the preferred orientation degree (hx) of the Ni(OH)2 nanoplates that is in good accordance with the value (hM) measured for the Ni(OH)2 nanoplates determined with the SEM and TEM images” the underlined text must be deleted.
I will insist that after equations 1, 2 and 4 here, for and here, respectively, must be written in capital letter because they belong to different sentences from the previous to each equation.
Literature has been substantially modified. In reference 9 appears “… Appl. Surf: Sci. Applied Surface Science…”, and the underlined words must be deleted.
I am convinced that with these modifications the article will be more attractive for readers and can be considered by the editor for publication.
Author Response
Thanks for your review. Your views are very important for improving this manuscipt. The revised parts are marked in blue.

Round 3
Reviewer 2 Report
I do not think the revised manuscript reflects my previous two comments at all. The paper contains three key values, i) preferred orientation, which was obtained from the fitting using March-Dollase function, in the powder packed in the XRD sample holder, ii) preferred growth of crystallites obtained by Žunić’s method, and iii) preferred growth of crystallites calculated from direct observation in TEM. The essence of the last two comments was: do not mix up the preferred orientation (i) and the preferred growth of crystallites (ii and iii). In the current revise, the authors just replaced the expression of i) from preferred “orientation” to preferred “growth”. Now this is simply incorrect. I cannot recommend the paper that neglects the context of current science.
Reviewer 3 Report
The manuscript has been improved after the suggested minor corrections. Now it is acceptable to be published by Nanomaterials.